# Double Viterbi: Weight Encoding for High Compression Ratio and Fast On-Chip Reconstruction for Deep Neural Network

**Daehyun Ahn[1], Dongsoo Lee[2] , Taesu Kim[1], Jae-Joon Kim[1]**
[1]POSTECH, Department of Creative IT Engineering, Korea, [2]Samsung Research, Korea
`daehyun.ahn@postech.ac.kr, dslee3@gmail.com,`
`{taesukim,jaejoon}@postech.ac.kr`

## Abstract

Weight pruning has been introduced as an efficient model compression technique. Even though pruning removes significant amount of weights in a network, memory requirement reduction was limited since conventional sparse matrix formats require significant amount of memory to store index-related information. Moreover, computations associated with such sparse matrix formats are slow because sequential sparse matrix decoding process does not utilize highly parallel computing systems efficiently. As an attempt to compress index information while keeping the decoding process parallelizable, Viterbi-based pruning was suggested. Decoding non-zero weights, however, is still sequential in Viterbi-based pruning. In this paper, we propose a new sparse matrix format in order to enable a highly parallel decoding process of the entire sparse matrix. The proposed sparse matrix is constructed by combining pruning and weight quantization. For the latest RNN models on PTB and WikiText-2 corpus, LSTM parameter storage requirement is compressed $19\times$ using the proposed sparse matrix format compared to the baseline model. Compressed weight and indices can be reconstructed into a dense matrix fast using Viterbi encoders. Simulation results show that the proposed scheme can feed parameters to processing elements **20 %** to **106 %** faster than the case where the dense matrix values directly come from DRAM.

## 1 Introduction

Deep neural networks (DNNs) require significant amounts of memory and computation as the number of training data and the complexity of task increases (Bengio & Lecun, 2007). To reduce the memory burden, pruning and quantization have been actively studied. Pruning removes redundant connections of DNNs without accuracy degradation (Han et al., 2015). The pruned results are usually stored in a sparse matrix format such as compressed sparse row (CSR) format or compressed sparse column (CSC) format, which consists of non-zero values and indices that represent the location of non-zeros. In the sparse matrix formats, the memory requirement for the indices is not negligible.

Viterbi-based pruning (Lee et al., 2018) significantly reduces the memory footprint of sparse matrix format by compressing the indices of sparse matrices using the Viterbi algorithm (Forney, 1973). Although Viterbi-based pruning compresses the index component considerably, weight compression can be further improved in two directions. First, the non-zero values in the sparse matrix can be compressed with quantization. Second, sparse-to-dense matrix conversion in Viterbi-based pruning is relatively slow because assigning non-zero values to the corresponding indices requires sequential processes while indices can be reconstructed in parallel using a Viterbi Decompressor (VD).

Various quantization techniques can be applied to compress the non-zero values, but they still cannot reconstruct the dense weight matrix quickly because it takes time to locate non-zero values to the corresponding locations in the dense matrix. These open questions motivate us to find a non-zero value compression method, which also allows parallel sparse-to-dense matrix construction. The contribution of this paper is as follows.

(a) To reduce the memory footprint of neural networks further, we propose to combine the Viterbi-based pruning (Lee et al., 2018) with a novel weight-encoding scheme, which also uses the Viterbi-based approach to encode the quantized non-zero values.

(b) We suggest two main properties of the weight matrix that increase the probability of finding "good" Viterbi encoded weights. First, the weight matrix with equal composition ratio of '0' and '1' for each bit is desired. Second, using the pruned parameters as "Don't Care" terms increases the probability of finding desired Viterbi weight encoding.

(c) We demonstrate that the proposed method can be applied to Recurrent Neural Networks (RNNs) and Convolutional Neural Networks (CNNs) with various sizes and depths.

(d) We show that using the same Viterbi-based approach to compress both indices and non-zero values allows us to build a highly parallel sparse-to-dense reconstruction architecture. Using a custom cycle-simulator, we demonstrate that the reconstruction can be done fast.

## 2 RELATED WORKS

DNNs have been growing bigger and deeper to solve complex nonlinear tasks. However, Denil et al. (2013) showed that most of the parameters in neural networks are redundant. To reduce the redundancy and minimize memory and computation overhead, several weight reduction methods have been suggested. Recently, magnitude-based pruning methods became popular due to its computational efficiency (Han et al., 2015). Magnitude-based pruning methods remove weights according to weight magnitude only and retrain the pruned network to recover from accuracy loss. The method is scalable to large and deep neural networks because of its low computation overhead. Han et al. (2015) showed $9\times$-$13\times$ pruning rate on AlexNet and VGG-16 networks without accuracy loss on ImageNet dataset. Although the compression rate was high, reduction of actual memory requirement was not as high as the compression rate because conventional sparse matrix formats, such as CSR and CSC, must use large portion of memory to store the indices of surviving weights. Lee et al. (2018) succeeded in reducing the amount of index-related information using a Viterbi-algorithm based pruning method and corresponding custom sparse matrix format. Lee et al. (2018) demonstrated 38.1% memory reduction compared to Han et al. (2015) with no accuracy loss. The memory reduction was limited, however, due to uncompressed non-zero values.

Several weight quantization methods were also suggested to compress the parameters of neural networks. Courbariaux et al. (2016); Li et al. (2016); Rastegari et al. (2016) demonstrated that reducing the weights to binary or ternary was possible, but the accuracy loss of the binary neural networks was significant. Zhou et al. (2016) reduced the bit resolution of weights to binary, activations to 2 bits and gradients to 6 bits with 9.8 % top-1 accuracy loss on AlexNet for ImageNet task. Guo et al. (2017) demonstrated a binary-weight AlexNet with 2.0% top-1 accuracy loss, achieving $\sim 10\times$ compression rate. Xu et al. (2018) showed that RNNs can also be quantized to reduce the memory footprint. By quantizing the weight values to 3 bits with proposed method, the memory footprint of RNN models were reduced $\sim 10.5\times$ with negligible performance degradation. Han et al. (2016b) suggested to combine pruning with weight quantization to achieve higher compression rate. The results showed $35\times$ increase in compression rate on AlexNet. However, the reduction was limited since the memory requirement of index-related information was only slightly improved with Huffman coding.

Although several magnitude-based pruning methods showed high compression rate, computation time did not improve much, because it takes time to decode the sparse matrix formats that describe irregular weight indices of pruned networks. Han et al. (2016a; 2017) suggested to use dedicated hardware, custom sparse matrix formats, and dedicated pruning methods to accelerate the computation even after pruning. Hanson & Pratt (1989); Yu et al. (2017) tried to accelerate the computation by limiting the irregularity of weight indices. By pruning neurons or feature maps, pruned weight matrices could maintain the dense format. These approaches successfully reduced the number of computation of neural networks, but the compression rate was limited due to additional pruning conditions. Although Lee et al. (2018) could use the Viterbi encoder to construct the index matrix fast, the process of pairing the non-zero weight values with the corresponding indices is still sequential, and thus relatively slow.

# 3 WEIGHT PRUNING AND QUANTIZATION USING DOUBLE-VITERBI APPROACH

Figure 1 illustrates the flowchart of the proposed compression method. Viterbi-based pruning (Lee et al., 2018) is applied first, and the pruned matrix is quantized using alternating multi-bit quantization method (Xu et al., 2018). Quantized binary code matrices are then encoded using the Viterbi-based approach, which is similar to the one used in pruning.

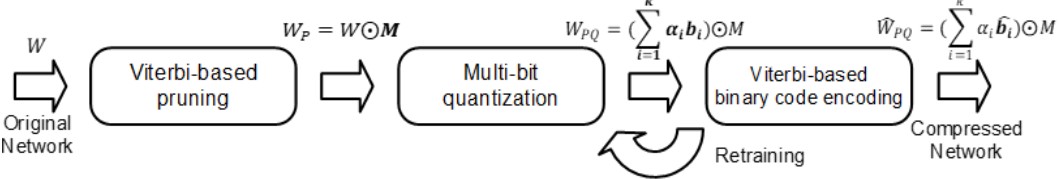

Figure 1: Flowchart of Double Viterbi compression. $W$ is the weight of an original network and $W_P$, $W_{PQ}$, and $\hat{W}_{PQ}$ represent the compressed weights after each process. $M \in \{0, 1\}$ is an index matrix which indicates whether each weight is pruned or not, and $\odot$ means element-wise multiplication. $\{\alpha_i\}_{i=1}^k \in \mathbb{R}$ and $\{\mathbf{b}_i\}_{i=1}^k \in \{-1, +1\}$ are constants and binary weights generated by quantization. $\{\hat{\mathbf{b}}_i\}_{i=1}^k \in \{-1, +1\}$ is binary weights encoded by the Viterbi algorithm.

## 3.1 VITERBI-BASED PRUNING FOR SPARSE MATRIX INDEX COMPRESSION

As the first step of the proposed weight encoding scheme, we compress the indices of the non-zero values in sparse weight matrix using the Viterbi-based pruning (Figure 1) (Lee et al., 2018). In this scheme, Viterbi algorithm is used to select a pruned index matrix which minimizes the accuracy degradation among many candidates which a Viterbi decompressor can generate. While the memory footprint of the index portion is significantly reduced by the Viterbi-based pruning, the remaining non-zero values after pruning still require non-negligible memory when high-precision bits are used. Hence, quantization of the non-zero values is required for further reduction of the memory requirement. Appendix A.1 explains the Viterbi-based pruning in detail.

## 3.2 MULTI-BIT QUANTIZATION AFTER VITERBI-BASED PRUNING

After Viterbi-based pruning is finished, the alternating multi-bit quantization (Xu et al., 2018) is applied to the sparse matrix (Figure 1). As suggested in Xu et al. (2018), real-valued non-zero weights are quantized into multiple binary codes $\{\mathbf{b}_i\}_{i=1}^k \in \{-1, +1\}$. Detailed algorithm is explained in Appendix A.2.

In addition to the high compression capabilities, another important reason we chose the alternating quantization is that the output distribution of the method is well suited to the Viterbi algorithm, which is used to encode the quantized non-zero values. Detailed explanation is given in Section 3.3.

## 3.3 ENCODING BINARY WEIGHT CODES USING THE VITERBI ALGORITHM

A sparse matrix that is generated by Viterbi-based pruning and quantization can be represented using the Viterbi Compression Matrix (VCM) format (Lee et al., 2018). A sparse matrix stored in VCM format requires much smaller amount of memory than the original dense weight matrix does. However, it is difficult to parallelize the process of reconstructing sparse matrix from the representation in VCM format, because assigning each non-zero value to its corresponding index requires a sequential process of counting ones in indices generated by the Viterbi encoder. To address this issue, we encode binary weight codes $\{\mathbf{b}_i\}_{i=1}^k$ as $\{\hat{\mathbf{b}}_i\}_{i=1}^k$ in addition to the indices, based on the same Viterbi algorithm (Forney, 1973). By using similar VD structures (Figure 2) to generate both $\{\hat{\mathbf{b}}_i\}_{i=1}^k$ and indices, we can generate both $\{\hat{\mathbf{b}}_i\}_{i=1}^k$ and corresponding indices at the same time; thereby parallel sparse-to-dense matrix conversion becomes possible as shown in Figure 3.

While using VD structures to generate binary weight codes allows parallel sparse-to-dense matrix conversion, it requires the quantization method to satisfy a specific condition to minimize accuracy

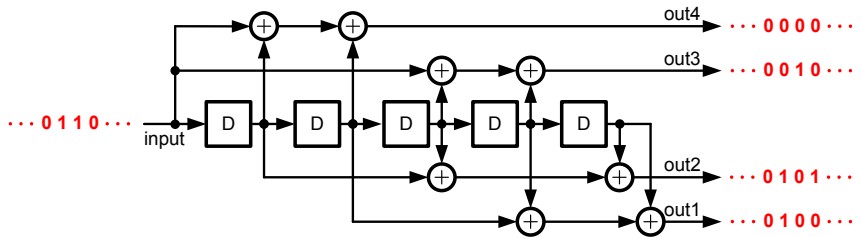

Figure 2: Structure of Viterbi decompressor (VD) which decodes a binary code compressed to 1/4 times. "D" indicates a D Flip-Flop which delays an input data for a clock cycle in the design and $\bigoplus$ indicates an XOR gate.

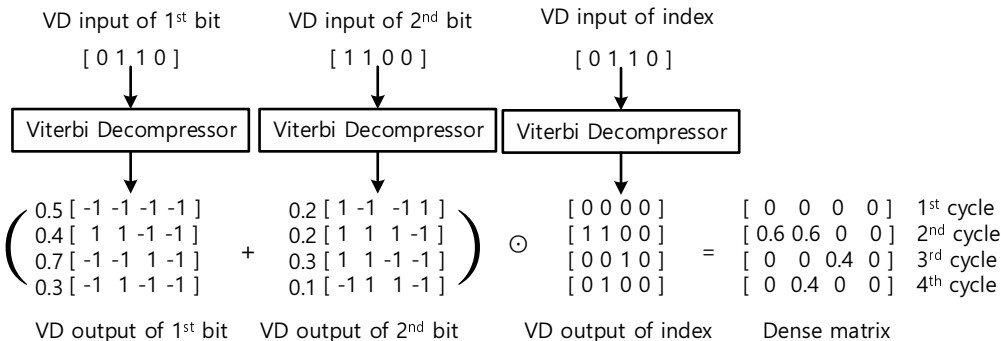

Figure 3: Proposed process of sparse-to-dense matrix conversion for the Viterbi-based compressed matrix. This figure shows an example such that weight values and weight index values are generated by independent Viterbi decompressors simultaneously.

loss after Viterbi-based encoding. It is known that the VD structure acts as a random number generator (Lee & Roy, 2012), which produces '0' and '1' with 50 % probability each. Thus, generated binary weight codes will be closer to the target binary weight codes if the target binary weight code matrix also consists of equal number of '0' and '1'. Interestingly, the composition ratio of '-1' and '+1' in each $\mathbf{b}_i$, which was generated by the alternating quantization method, is 50 % each.

It is because the weights in DNNs are generally initialized symmetrically with respect to '0' (Glorot & Bengio, 2010; He et al., 2015) and the distribution is maintained even after training (Lin et al., 2016). The preferable output distribution of the alternating quantization implies that the probability of finding an output matrix $\hat{\mathbf{b}}_i$ close to $\mathbf{b}_i$ with the Viterbi algorithm is high. For comparison, we measured the accuracy differences before and after Viterbi encoding for several quantization methods such as linear quantization (Lin et al., 2016), logarithmic quantization (Miyashita et al., 2016), and alternating quantization (Xu et al., 2018). When the Viterbi encoding is applied to the weight quantized by alternating quantization (Xu et al., 2018), the validation accuracy degrades by only 2 %. However, accuracy degrades by 71 % when the Viterbi encoding is applied to the weight quantized using other methods (Lin et al., 2016; Miyashita et al., 2016). The accuracy difference mainly comes from the uneven weight distribution. Because weights of neural networks usually have normal distribution, the composition ratio of '0' and '1' is not equal when the linear or logarithmic quantization is applied to the weights unlike alternating quantization.

Another important idea to increase the probability of finding "good" Viterbi encoded weight is to consider the pruned parameters in $\mathbf{b}_i$ as "Don't Care" terms (Figure 4). The "Don't Care" elements can have any values when finding $\hat{\mathbf{b}}_i$, because they will be masked by the zero values in the index matrix generated by the Viterbi pruning.

Next, let us describe how we use the Viterbi algorithm for weight encoding. We select the $\hat{\mathbf{b}}_i$ that best matches with $\mathbf{b}_i$ among all possible $\hat{\mathbf{b}}_i$ cases that the VD can generate, as follows. We first construct a trellis diagram as shown in Figure 5. The trellis diagram is a state diagram represented

**(a)** $\begin{bmatrix} \mathbf{x} & \mathbf{x} & \mathbf{x} & \underline{\mathbf{1}} & \mathbf{x} & \mathbf{x} \end{bmatrix}$
$\begin{bmatrix} \underline{-\mathbf{1}} & \mathbf{x} & \underline{\mathbf{1}} & \mathbf{x} & \mathbf{x} & \mathbf{x} \end{bmatrix}$
$\begin{bmatrix} \mathbf{x} & \mathbf{x} & \mathbf{x} & \mathbf{x} & \underline{-\mathbf{1}} & \mathbf{x} \end{bmatrix}$
$\begin{bmatrix} \mathbf{x} & \mathbf{x} & \mathbf{x} & \underline{\mathbf{1}} & \mathbf{x} & \mathbf{x} \end{bmatrix}$

**(b)** $\begin{bmatrix} \mathbf{1} & \mathbf{1} & \mathbf{1} & \underline{\mathbf{1}} & \mathbf{1} & \mathbf{1} \end{bmatrix}$
$\begin{bmatrix} \underline{\mathbf{0}} & \mathbf{0} & \underline{\mathbf{1}} & \mathbf{1} & \mathbf{1} & \mathbf{1} \end{bmatrix}$
$\begin{bmatrix} \mathbf{0} & \mathbf{0} & \mathbf{0} & \mathbf{0} & \underline{\mathbf{0}} & \mathbf{0} \end{bmatrix}$
$\begin{bmatrix} \mathbf{1} & \mathbf{1} & \mathbf{1} & \underline{\mathbf{1}} & \mathbf{1} & \mathbf{1} \end{bmatrix}$

Figure 4: (a) Target binary weight code $\mathbf{b}_i$ and (b) corresponding $\hat{\mathbf{b}}_i$ generated from VD. $x$ indicates a pruned "Don't care" term, and elements with underline indicate surviving weights. '0' in (b) corresponds to '-1' in (a).

with time index $T$. A state number is represented with Flip-Flop (FF) values, where the rightmost FF value is the most significant bit (MSB).[1] Each transition with a 1-bit input from a state generates the corresponding multiple output bits.

A cost function for each transition using path and branch metrics is set and computed in the next step. The branch metric $\lambda_t^{i,j}$ is the cost of traveling along a transition from a state $i$ to the successor state $j$ at the time index $t$. The path metric is expressed as

$$\Gamma_{t+1}^j = \max\left(\Gamma_t^{i1} + \lambda_t^{i1,j}, \Gamma_t^{i2} + \lambda_t^{i2,j}\right), \tag{1}$$

where $i1$ and $i2$ are two predecessor states of $j$. Equation 1 denotes that one of the two possible transitions is selected to maximize the accumulated value of branch metrics.[2] The branch metric is defined as

$$\beta_t^{i,j,m} = \begin{cases} 1, & \text{if } b_t^{i,j,m} = 2o_t^{i,j,m} - 1 \\ 0, & \text{otherwise} \end{cases}, \quad \lambda_t^{i,j} = \sum_{m=1}^{N_o} \beta_t^{i,j,m}, \tag{2}$$

where $b_t^{i,j,m}$ is the value of binary codes according to the $m^{th}$ VD output at time index $t$, $o_t^{i,j,m}$ is the value of the $m^{th}$ VD output at time index $t$, and $N_o$ is the number of outputs generated by the VD at each time step. $\{-1, +1\}$ in the binary codes corresponds to $\{0, 1\}$ in the VD output in equation 2. Equation 2 maximizes the number of VD outputs that exactly match with corresponding binary codes while ignoring the pruned parameters (Figure 4). When the last time index is reached, the state with the maximum path metric is selected. Previous states connected by surviving branches are traced while corresponding $o_t^{i,j,m}$ of each branch is recorded as $\hat{b}_t^{i,j,m}$. Each binary weight code $\mathbf{b}_i$ is encoded as $\hat{\mathbf{b}}_i$ with a compression ratio of $1/N_o$ using this scheme.

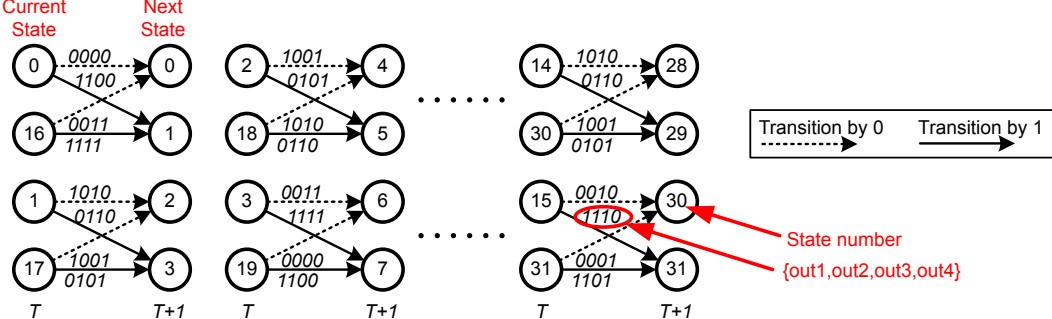

Figure 5: Trellis diagram of the VD in Figure 2. Each circle indicates a state. A circle which is the source point of arrows indicates a current state and a circuit which is the sink point of arrows indicates a next state. The arrow indicates a transition from the current state to the next state. Depending on the input to VD, each current state can be switched to one of the two potential next states in the next clock. The number in a circle indicates the index for the state.

---

[1] If the VD has $N$ FFs, then the number of states is $2^N$.

[2] max function is used instead of min function because the metric value is considered as a degree of 'reward' rather than 'cost' (Lee et al., 2018).

## 3.4 RETRAINING

To maintain the accuracy, the number of incorrect bits in the encoded binary code $\hat{\mathbf{b}}_i$ compared to the original binary code $\mathbf{b}_i$ needs to be minimized. Thus, we retrain the network with $\hat{W}_{PQ} = \left(\sum_{i=1}^{k} \alpha_i \hat{\mathbf{b}}_i\right) \odot \mathbf{M}$ ($\mathbf{M}$ is the index matrix of non-zeros in $\mathbf{W}$), apply the alternating quantization, and then perform the Viterbi encoding repeatedly (Figure 1). By repeating the retraining, quantization, and Viterbi encoding, the number of incorrect bits between $\hat{\mathbf{b}}_i$ and $\mathbf{b}_i$ can be reduced because the parameters in the network are fine-tuned close to parameters in $\hat{W}_{PQ}$. During the retraining period, we apply the straight-through estimate (Rastegari et al., 2016), i.e. $\frac{\partial C}{\partial \hat{W}_{PQ}} = \frac{\partial C}{\partial W}$ as adopted in Xu et al. (2018). After the last Viterbi encoding is finished, small amount of components in $\hat{\mathbf{b}}_i$ can be still different from the corresponding values in $\mathbf{b}_i$. To maintain the accuracy, location data for the incorrect components are stored separately and are used to flip the corresponding VD encoded bits during on-chip weight reconstruction period. In our experiments, the memory requirement for the correction data was negligible. After the retraining is finished, we can obtain a compressed parameter in Viterbi Weight Matrix (VWM) format, which includes $\{\alpha_i\}_{i=1}^{k}$, compressed input to generate $\{\hat{\mathbf{b}}_i\}_{i=1}^{k}$, compressed index in VCM format, and indices where $\{\hat{\mathbf{b}}_i\}_{i=1}^{k} \neq \{\mathbf{b}_i\}_{i=1}^{k}$. Note that entire training process used the training dataset and the validation dataset only to decide the best compressed weight data. The accuracy measurement for the test dataset was done only after training is finished so that any hyperparameter was not tuned on the test dataset. All the experiments in this paper followed the above training principle.

## 4 EXPERIMENTAL RESULTS

### 4.1 RECURRENT NEURAL NETWORKS (RNN) FOR LANGUAGE MODELING

We first conduct experiments on Penn Tree Bank (PTB) corpus (Marcus et al., 1993). We use the standard split version of PTB corpus with 10K vocabulary (Mikolov, 2012), and evaluate the performance using perplexity per word (PPW). We pretrain the RNN model[3] which contains 1 layer of LSTM with 600 memory units, then prune the parameters of LSTMs with 80 % pruning rate using the Viterbi-based pruning technique[4] and retrain the model. Then, we quantize the parameters of LSTMs using alternating quantization technique, encode the binary weight codes by using the Viterbi algorithm, and retrain the model. We repeat the quantization, binary code encoding, and retraining process 5 times.

**Number of quantization bits:** We quantize the LSTM model with different numbers of quantization bits $k$ with the fixed $N_o = 5$. As $k$ increases, PPW is improved, but the memory requirement for parameters is also increased (Table 1). Note that $k = 3$ is the minimum number of bits that minimizes the model size without PPW degradation. Compared to Lee et al. (2018), further quantization and Viterbi-based compression reduce the parameter size by 78 % to 90 % (Table 1).

**Number of VD outputs:** We compress the binary weight codes with different number of VD outputs $N_o$ in case of $k = 3$. As $N_o$ increases, PPW degrades while the memory requirement for parameters is increased (Table 1). Large $N_o$ implies that the binary weight codes are compressed with high compression ratio $1/N_o$, but the similarity between $\hat{\mathbf{b}}_i$ and $\mathbf{b}_i$ decreases. The optimal $N_o$ is 100/(100-pruning rate (%)) , where the average number of survived parameters per $N_o$ serial parameters is 1 statistically, which results in no model performance degradation.

**Effectiveness of "Don't Care":** To verify the effectiveness of using the "Don't Care" elements, we apply our proposed method on the original network and pruned one. While the pruned network maintains the original PPW after applying our proposed compression method, applying our method to the dense network degrades PPW to 102.6. This is because the ratio of incorrect bits between $\hat{\mathbf{b}}_i$ and $\mathbf{b}_i$ decreases from 28.3 % to 1.7 % when we use the sparse $\mathbf{b}_i$. Therefore, combination of the Viterbi pruning and alternating quantization increases the probability of finding $\hat{\mathbf{b}}_i$ close to $\mathbf{b}_i$ using the VD for weight encoding, which results in no PPW degradation.

---

[3]https://github.com/tensorflow/models/blob/master/tutorials/rnn/ptb/ptb_word_lm.py

[4]We use the VD which consists of a 50-bit VD output and a 5-bit comparator input. 1 skip state is applied to the Viterbi algorithm (Lee et al., 2018).

Table 1: Compression result of LSTM model on the PTB corpus with different $k$ and $N_o$.

| $k$ | $N_o$ | PPW | Parameter size | Compression rate |
|---|---|---|---|---|
| 2 | 5 | 84.9 | 234.4 KB | 2.1 % (48×) |
| 3 | 5 | **84.6** | 337.5 KB | **3.0 % (33×)** |
| 4 | 5 | 84.1 | 502.2 KB | 4.5 % (22×) |
| 3 | 2 | 84.6 | 611.2 KB | 5.4 % (18×) |
| 3 | 5 | **84.6** | 337.5 KB | **3.0 % (33×)** |
| 3 | 10 | 85.3 | 290.5 KB | 2.6 % (39×) |
| Pruning (Lee et al., 2018) | | 84.6 | 2320.3 KB | 20.6 % (5×) |
| Baseline (Uncompressed) | | 84.6 | 11250.0 KB | |

**The latest RNN for language modeling:** We further test our proposed method on the latest RNN model (Yang et al., 2018), which shows the best perplexity on both PTB and WikiText-2 (WT2) corpus. We prune 75 % of the parameters in three LSTMs with the same condition as we prune the above 1-layer LSTM model, and quantize them to 3 bits ($k = 3$). Note that we do not apply fine-tuning and dynamic evaluation (Krause et al., 2017) in this experiment. The compression result in Table 2 shows that the memory requirements for the models are reduced by **94.7 %** with our VWM format on both PTB and WT2 corpus without PPW degradation. This result implies that our proposed compression method can be applied regardless of the depth and size of the network. Detailed experiment settings and compression results are described in Appendix A.3. In addition, we extend our proposed method to the RNN models for machine translation (Wu et al., 2016), and its experimental results are presented in Appendix A.4.

Table 2: Compression result of the lastest LSTM models for PTB and WT2 corpus.

| Corpus | Compression Scheme | LSTM Parameter Size | Compression Rate | Validation PPW | Test PPW |
|---|---|---|---|---|---|
| PTB | Baseline (Uncompressed) | 62706.3 KB | - | 58.7 | 56.3 |
| | Lee et al. (2018) | 16038.4 KB | 25.6 % (4×) | 59.4 | 56.6 |
| | VWM (Ours) | 3299.5 KB | **5.3 % (19×)** | **58.7** | **56.2** |
| WT2 | Baseline (Uncompressed) | 85664.1 KB | - | 67.0 | 64.0 |
| | Lee et al. (2018) | 22067.8 KB | 25.8 % (4×) | 67.6 | 64.6 |
| | VWM (Ours) | 6732.3 KB | **5.3 % (19×)** | **67.5** | **64.4** |

## 4.2 CONVOLUTIONAL NEURAL NETWORKS (CNN) FOR IMAGE CLASSIFICATION

We also apply our proposed method to a CNN, VGG-9 (2×128C3 - 2×256C3 - 2×512C3 - 2×1024FC - 10SM[5]) on CIFAR-10 dataset to verify the proposed technique is valid for other types of DNNs. We randomly select 5 K validation images among 50 K training images in order to observe validation error during retraining process and measure the test error after retraining. We use $k = 3$ for all layers. Optimal $N_o$ for each layer is chosen based on the pruning rate of the parameters; $N_o = 4$ for convolutional layers, $N_o = 25$ for the first two fully-connected layers, and $N_o = 5$ for the last fully-connected layer. We also compute the memory requirement for other compression methods.

Experimental results on VGG-9 is found in Table 3. Compared to Han et al. (2016b), the VWM format generated by the proposed scheme has **39 %** smaller memory footprint due to the compressed indices, smaller number of bits for quantization, and encoded binary weight codes. This experiment on CIFAR-10 shows that our proposed method can be applied to DNNs with various types and sizes. Meanwhile, it can be seen that the combination of the Viterbi-pruning (Lee et al. (2018)) with the alternating quantization (Xu et al. (2018)) requires 10% smaller memory requirement than the VWM

---

[5]$nCm$ means a convolution layer where the number of output channel is $n$ with $mxm$ size of kernel. MP2 means a max-pooling layer with 2x2 size of kernel. $n$FC is a fully-connected layer with $n$ output neurons, and 10SM is a softmax layer with 10 labels.

format because the VWM format requires additional memory for indices where $\{\hat{\mathbf{b}}_i\}_{i=1}^k \neq \{\mathbf{b}_i\}_{i=1}^k$. However, additional "Viterbi-based binary code encoding" process for the VWM format allows parallel sparse-to-dense matrix conversion, which increases the parameter feeding rate up to 40.5 % compared to Lee et al. (2018). In Section 4.3, we analyze the speed of sparse-to-dense matrix conversion in detail.

Table 3: VGG-9 compression result on CIFAR-10 dataset.

| Layer | Parameter Size (KB) | Pruning Rate (%) | Compression rate (%) | | | | |
|---|---|---|---|---|---|---|---|
| | | | Pruning[a)] | | Pruning + Quantization | | |
| | | | | | Han et al. (2016b) | **Proposed** | |
| | | | Han et al. (2015) | Lee et al. (2018) | | Lee et al. (2018) +Xu et al. (2018) | VWM |
| Conv1[c)] | 13.5 | - | - | - | 25.0 | 11.2 | 11.2 |
| Conv2 | 576.0 | 74.6 | 50.9 | 28.5 | 10.3 | 5.5 | 6.0 |
| Conv3 | 1152.0 | 75.2 | 49.6 | 27.9 | 10.1 | 5.5 | 6.0 |
| Conv4 | 2304.0 | 75.0 | 50.1 | 28.2 | 10.2 | 5.5 | 6.0 |
| Conv5 | 4608.0 | 74.8 | 50.0 | 28.1 | 10.2 | 5.5 | 6.0 |
| Conv6 | 9216.0 | 75.4 | 50.5 | 28.4 | 10.3 | 5.5 | 6.0 |
| Fc1 | 32768.0 | 96.0 | 8.1 | 4.7 | 1.6 | 1.0 | 1.1 |
| Fc2 | 4096.0 | 95.9 | 8.3 | 4.7 | 1.7 | 1.0 | 1.1 |
| Fc3 | 40.0 | 79.7 | 40.8 | 23.5 | 8.3 | 5.0 | 5.5 |
| Total | 54733.5 | 89.2 | **21.9** (**5×**) | **12.2** (**8×**) | **4.5** (**22×**) | **2.5** (**40×**) | **2.7** (**36×**) |
| Validation error (Uncompressed model:11.5 %) | | | 11.3 % | 11.4 % | 11.2 % | 11.2 % | 11.4 % |
| Test error (Uncompressed model:12.2 %) | | | 12.2 % | 12.2 % | 12.2 % | 12.2 % | 12.4 % |

$a$) Non-zero values are represented as 32-bit floating point numbers.
$b$) Convolution filters are quantized to 8-bit, and weights of fully-connected layers and indices of sparse matrices are quantized to 5-bit, which is the same quantization condition as the condition used in Han et al. (2016b).
$c$) For the Conv1 layer, pruning is not applied and only the alternating quantization is applied.

## 4.3 ANALYSIS ON RECONSTRUCTION SPEED

We built a cycle-level simulator for the weight matrix reconstruction process of the proposed format to show that the sparse matrix-matrix multiplications with the proposed method can be done fast with parallel reconstruction of dense matrix. In the simulator, baseline structure feeds two dense input matrices to processing elements (PEs) using raw data fed by DRAM (Figure 6a), while the proposed structure reconstructs both index masks and binary codes using the highly compressed data fed by DRAM and sends the reconstructed values to PEs (Figure 6b). Both index masks and binary codes are reconstructed by several Viterbi encoders in parallel, and bit errors in binary codes are corrected in a serial manner using the small number of flip-bit related data, which are received from DRAM. Simulation results show that the feeding rate of the proposed scheme is 20.0-106.4 % higher than the baseline case and 10.3-40.5 % higher than Lee et al. (2018), depending on the pruning rate (Figure 6c). The gain mainly comes from the high compression rate and parallel reconstruction process of the proposed method. As shown in Figure 6c, higher sparsity leads to higher feeding rate. Higher sparsity allows using many VD outputs for the index $N_{ind}$, and increasing $N_{ind}$ leads to faster reconstruction. Also, the reconstruction rate of binary codes becomes higher with reduced number of non-zero values and corresponding bit corrections.

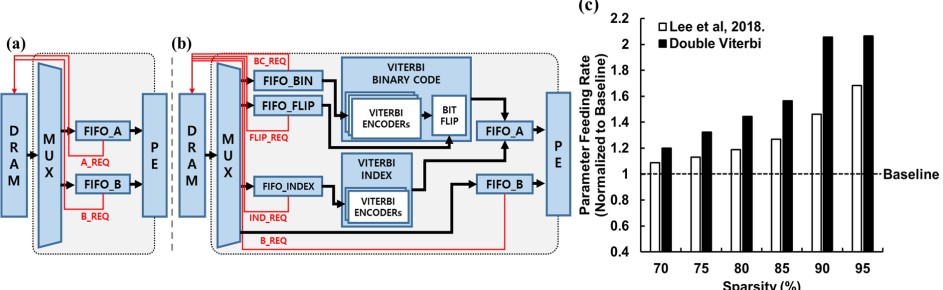

Figure 6: (a) Simplified diagrams of baseline and (b) Viterbi-based computation architectures. (c) Rate of parameter feeding into PEs for the proposed scheme compared to those for the baseline structure, which receives the dense matrix data directly from DRAM, and Lee et al. (2018). We assumed the number of VD outputs for the index $N_{ind} = 3, 4, 5, 6, 10, 10$ respectively as the reciprocal of each sparsity value. We used $N_{ind} = 10$ for 95 % sparsity since we compressed matrices with over 90 % sparsity with $N_{ind} = 10$. We also assumed $k = 3$, and 1 % bit-wise difference between $\hat{\mathbf{b}}_i$ and $\mathbf{b}_i$ during simulation. We also assumed that 16 non-zero parameters can be fed into the PE array in parallel and DRAM requires 10 cycles to handle a 256 bit READ operation.

## 5 CONCLUSIONS

We proposed a DNN model compression technique with high compression rate and fast dense matrix reconstruction process. We adopted the Viterbi-based pruning and alternating multi-bit quantization technique to reduce the memory requirement for both non-zeros and indices of sparse matrices. Then, we encoded the quantized binary weight codes using Viterbi algorithm once more. As the non-zero values and the corresponding indices are generated in parallel by multiple Viterbi encoders, the sparse-to-dense matrix conversion can be done very fast. We also demonstrated that the proposed scheme significantly reduces the memory requirements of the parameters for both RNN and CNN.

## ACKNOWLEDGMENTS

This research was in part supported by Samsung Research Funding Center of Samsung Electronics under Project Number SRFC-TC1603-04 and the MSIT(Ministry of Science and ICT), Korea, under the ICT Consilience Creative program(IITP-2018-2011-1-00783) supervised by the IITP(Institute for Information & Communications Technology Promotion).

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

# A   APPENDIX

## A.1   PRUNING USING THE VITERBI ALGORITHM

In Viterbi-based pruning scheme, the binary outputs generated by a Viterbi Decompressor (VD) are used as the index matrix that indicates whether a weight element is pruned ('0') or not ('1'). Suppose the number of elements in a target weight matrix is $q$, and the number of outputs generated by a VD at each time step is $N_{ind}$, then only $2^{q/N_{ind}}$ binary matrices can be generated by the VD among all $2^q$ binary matrices. The index matrix which minimizes the accuracy loss should be selected among binary matrix candidates which VD can generate in this pruning scheme, and the Viterbi algorithm is used for this purpose.

The overall pruning process is similar to the binary weight encoding process using the Viterbi algorithm in Section 3.3. First, Trellis diagram (Figure 5) of the VD which is used for pruning is constructed, and then the cost function is computed by using the path metric and the branch metric. The same path metric shown in Equation 1 in Section 3.3 is used to select the branch which maximizes the path metric between two connected branches from the previous states. On the other hand, a different branch metric $\lambda_t^{i,j}$ is used for pruning, which is expressed as:

$$D_t^{i,j,m} = \left( W_t^{i,j,m} - TH_p \right)/S_1,\, 0 \leq W_t^{i,j,m}, TH_p \leq 1$$

$$\beta_t^{i,j,m} = \begin{cases} \tanh\left( D_t^{i,j} \right) \times S_2, \text{when survived} \\ -\tanh\left( D_t^{i,j} \right) \times S_2, \text{when pruned} \end{cases} ,\, \lambda_t^{i,j} = \sum_{m=1}^{R} \beta_t^{i,j,m}, \tag{3}$$

where $W_t^{i,j,m}$ is the magnitude of a parameter at the $m^{\text{th}}$ VD output and time index $t$, normalized by the maximum absolute value of all elements in target weight matrix, and $TH_p$ is the pruning threshold value determined heuristically. As $\beta_t^{i,j,m}$ gives additional points (penalties) to the parameters with large magnitude to survive (be pruned), the possibility to prune small-magnitude parameters is maximized. $S_1$ and $S_2$ are the scaling factors which is empirically determined. (Lee et al. (2018) uses 5.0 and $10^4$ each). After computing the cost function through the whole time steps, the state with the maximum path metric is chosen, and we trace the previous state by selecting the surviving branch and corresponding indices backward until the first state is reached.

The ideal pruning rate of the Viterbi-based pruning is 50 %, because the VD structures act like random number generator and the probability to generate '0' or '1' is 50 % each. For various pruning rates, comparators and comparator threshold value, $TH_c$, are used. A $N_C$-bit comparator receives $N_c$ VD outputs and generates 1-bit result whether the value made by the combination of the received VD outputs (e.g. $\{out_1, out_2, \cdots, out_{N_{ind}}\}$ where $out_i$ indicates the $i_{th}$ VD output) is greater than $TH_c$ or not. For example, suppose a 4-bit comparator is used to the VD in Figure 1 and $TH_c = 3$, then the probability for the comparator to generate '1' is $25\%(= (3+1)/2^4)$ and this percentage is the target pruning rate. Comparators and $TH_c$ control the value of pruning rates and the index compression ratio decreases by $1/N_c$ times.

It is reported that a low $N_{ind}$ is desired to prune weights of convolutional layers while high $N_{ind}$ can be used to prune the weights of fully-connected layers because of the trade-off between the index compression ratio and the accuracy (Lee et al., 2018). Thus, in our paper, we use $N_{ind} = 50$ and $N_c = 5$ to prune weights of LSTMs and fully-connected layers in VGG-6 on CIFAR-10. On the other hand, we use $N_{ind} = 10$ and $N_c = 5$ to prune weights of convolutional layers in VGG-6 on CIFAR-10.

## A.2 ALTERNATING MULTI-BIT QUANTIZATION ON SPARSE MATRIX

---

**Algorithm 1:** Alternating multi-bit quantization for sparse weight

---

**input** : Sparse weight $\mathbf{w} \in \mathbb{R}^n$ , the number of quantization bits $k$,
the number of iterations $T$
**output** : $\{\alpha_i, \mathbf{b}_i\}_{i=1}^k$ , with $\alpha_i \in \mathbb{R}$, $\mathbf{b}_i \in \{-1, +1\}^n$
$\mathbf{r}_0 = \mathbf{w}, \mathbf{m} = sign(|\mathbf{w}|)$

> Step 1. Initialization

**for** $i \leftarrow 1$ *to* $k$ **do**

    $\alpha_i \leftarrow \frac{\sum |\mathbf{r}_{i-1}|}{\sum m}$ ;                > Average norm-1 of non-zeros

    $\mathbf{b}_i \leftarrow sign(\mathbf{r}_i)$ ;

    $\mathbf{r}_{i-1} \leftarrow \mathbf{w} - \sum_{j=1}^i \alpha_j \mathbf{b}_j$ ;

**end**
$\mathbf{B} \leftarrow [\mathbf{b}_1, \cdots, \mathbf{b}_k]$ ;

> Step 2. Alternating quantization

**if** $det(\mathbf{B}^T \mathbf{B}) \neq 0$ **then**

    **for** $t \leftarrow 1$ *to* $T$ **do**

        $[\alpha_1, \alpha_2, ..., \alpha_k] \leftarrow \left( \left( \mathbf{B}^T \mathbf{B} \right)^{-1} \mathbf{B}^T \mathbf{w} \right)^T$ ;

        Construct $\mathbf{v}$ in ascending order ;

        Update $\{\mathbf{b}_i\}_{i=1}^k$ according to $\mathbf{v}$ ; > Pruned components in $\{\mathbf{b}_i\}_{i=1}^k$ are set to '0'

        $\mathbf{B} \leftarrow [\mathbf{b}_1, \cdots, \mathbf{b}_k]$ ;

    **end**

**end**
$\mathbf{B} \leftarrow [\mathbf{b}_1 + (\neg m), \cdots, \mathbf{b}_k + (\neg m)]$ ;
> Pruned components in $\{\mathbf{b}_i\}_{i=1}^k$ are set to '+1' because $\mathbf{b}_i \in \{-1, +1\}^n$.

---

Algorithm 1 explains the multi-bit quantization process applied to a sparse matrix. Given $\{\alpha_i\}_{i=1}^k$ with $\alpha_1 \geq \alpha_2 \cdots \geq \alpha_{k-1} \geq \alpha_k \geq 0$ and the number of quantization bits $k$, each non-zero value in a sparse weight matrix $\mathbf{w}$ is quantized to a value $v \in \mathbf{v} = \{-\sum_{i=1}^k \alpha_i, -\sum_{i-1}^{k-1} \alpha_i + \alpha_k, \cdots, \sum_{i-1}^{k-1} \alpha_i - \alpha_k, \sum_{i=1}^k \alpha_i\}$. Algorithm 1 is a derivative of the alternating multi-bit quantization algorithm (Xu et al., 2018) with some consideration for sparse matrix. First, $\{\alpha_i\}_{i=1}^k$ is initialized with the average norm-1 value of non-zeros in $\mathbf{w}$ instead of the average norm-1 value of entire elements in a dense matrix, which was the case in Xu et al. (2018). Also, if the inverse matrix of $\mathbf{B}^T \mathbf{B}$ does not exist due to high pruning rate, Algorithm 1 does not proceed to the second alternating quantization step. Note that the model performance was not degraded much without running the second quantization step. Pruned components are represented with '0' during the quantization. After the quantization, $\{\mathbf{b}_i\}_{i=1}^k$ for the pruned components are set to '+1' because $\mathbf{b}_i \in \{-1, +1\}^n$. It does not matter whether pruned components are represented as '-1' or '+1', because they will be eventually masked by the binary index matrix indicating the location of non-zeros.

A.3 DETAILED COMPRESSION RESULTS OF THE LATEST RNN FOR LANGUAGE MODELING

The RNN model in Yang et al. (2018) is composed of three LSTM layers, and use various learning techniques such as mixture-of-softmaxes (MoS) to achieve better perplexity. As shown in Table A1 and Table A2, the parameters in the first layer have high sparsity, so we use $N_o = 6$. In the remaining layers, however, we use $N_o = 3$ because the parameters are pruned with only about 70 % pruning rate. We repeat the process of quantization, binary code encoding, and retraining only once.

Table A1: Compression result of the lastest LSTM model for PTB corpus.

| Layer | LSTM Parameter | Pruning | Compression rate (%) | |
| --- | --- | --- | --- | --- |
| | Size (KB) | Rate (%) | Lee et al. (2018) | VWM (ours) |
| LSTM1 | 18600.0 | 86.7 | 13.9 | 3.7 |
| LSTM2 | 28800.0 | 70.1 | 30.5 | 5.9 |
| LSTM3 | 15306.3 | 70.2 | 30.4 | 6.0 |
| Total | 62706.3 | 75.0 | **25.6** (**4×**) | **5.3** (**19×**) |
| Validation PPW | 58.7 | 58.7 | 59.4 | 58.7 |
| Test PPW | 56.3 | 56.3 | 56.6 | 56.2 |

Table A2: Compression result of the lastest LSTM model for WT2 corpus.

| Layer | LSTM Parameter | Pruning | Compression rate (%) | |
| --- | --- | --- | --- | --- |
| | Size (KB) | Rate (%) | Lee et al. (2018) | VWM (ours) |
| LSTM1 | 26054.7 | 86.1 | 14.5 | 3.7 |
| LSTM2 | 41328.1 | 69.4 | 31.3 | 6.0 |
| LSTM3 | 18281.3 | 71.2 | 29.4 | 5.9 |
| Total | 85664.1 | 74.9 | **25.8** (**4×**) | **5.3** (**19×**) |
| Validation PPW | 67.0 | 67.0 | 67.6 | 67.5 |
| Test PPW | 64.0 | 64.0 | 64.6 | 64.4 |

A.4 RECURRENT NEURAL NETWORKS (RNN) FOR MACHINE TRANSLATION

We also extend our experiments on the RNN models for machine translation (Wu et al., 2016) [6]. We use the model which consists of an encoder, a decoder and an attention layer. 4-layer LSTMs with 1024 units compose each encoder and decoder. A bidirectional LSTM (BiLSTM) is used for the first layer of the encoder. The weights of LSTM models are pruned with 75 % pruning rate by the Viterbi-based pruning techinque, then $k = 4$ is used for quantization. Optimal $N_o$ values are used according to the sparsity of each LSTM layer (i.e. $3 \leq N_o \leq 6$ is enough to encode binary weight codes with 70 - 83% of sparsity). The process of quantization, binary code encoding, and retraining is repeated only once in this case, too. As shown in Table A3 and Table A4, we reduce the memory requirement of each baseline model by **93.5 %** using our proposed technique. This experiment results show that our proposed scheme can be extended to RNNs for other complex tasks.

---

[6]https://github.com/tensorflow/nmt

Table A3: Compression result of the GNMT model for WMT En → De.

| Network Type | Layer | LSTM Parameter Size (KB) | Pruning Rate (%) | Compression rate (%) | |
|---|---|---|---|---|---|
| | | | | Lee et al. (2018) | VWM (ours) |
| Encoder | BiLSTM (FW) | 32768.0 | 83.3 | 17.3 | 4.8 |
| | BiLSTM (BW) | 32768.0 | 76.7 | 23.9 | 5.8 |
| | LSTM1 | 49152.0 | 74.1 | 26.5 | 6.2 |
| | LSTM2 | 32768.0 | 70.9 | 29.8 | 7.3 |
| | LSTM3 | 32768.0 | 74.2 | 26.4 | 6.3 |
| Decoder | LSTM4 | 49152.0 | 76.1 | 24.6 | 6.0 |
| | LSTM5 | 49152.0 | 79.3 | 21.3 | 5.7 |
| | LSTM6 | 49152.0 | 73.4 | 27.2 | 6.6 |
| | LSTM7 | 49152.0 | 69.3 | 31.3 | 7.8 |
| Total | | 376832.0 | 75.1 | **25.6** (**4×**) | **6.3** (**16×**) |
| Validation BLEU (WMT 15) | | | 25.6 | 25.8 | 25.2 |
| Test BLEU (WMT 16) | | | 30.1 | 29.9 | 29.0 |

Table A4: Compression result of the GNMT model for WMT De → En.

| Network Type | Layer | LSTM Parameter Size (KB) | Pruning Rate (%) | Compression rate (%) | |
|---|---|---|---|---|---|
| | | | | Lee et al. (2018) | VWM (ours) |
| Encoder | BiLSTM (FW) | 32768.0 | 81.6 | 19.0 | 5.0 |
| | BiLSTM (BW) | 32768.0 | 75.5 | 25.1 | 6.0 |
| | LSTM1 | 49152.0 | 73.9 | 26.7 | 6.3 |
| | LSTM2 | 32768.0 | 71.3 | 29.4 | 6.8 |
| | LSTM3 | 32768.0 | 73.4 | 27.2 | 6.5 |
| Decoder | LSTM4 | 49152.0 | 76.5 | 24.1 | 6.0 |
| | LSTM5 | 49152.0 | 80.3 | 20.3 | 5.5 |
| | LSTM6 | 49152.0 | 73.2 | 27.5 | 6.6 |
| | LSTM7 | 49152.0 | 68.9 | 31.8 | 7.8 |
| Total | | 376832.0 | 74.9 | **25.8** (**4×**) | **6.3** (**16×**) |
| Validation BLEU (WMT 15) | | | 28.0 | 28.4 | 28.0 |
| Test BLEU (WMT 16) | | | 33.2 | 33.3 | 33.0 |

