# OpenReview forum: "Double Viterbi: Weight Encoding for High Compression Ratio and Fast On-Chip Reconstruction for Deep Neural Network"
_ICLR.cc/2019/Conference_

### Official Review · AnonReviewer3 · 2018-11-01
**Hard to read, but idea is interesting**

**Rating:** 7
**Confidence:** 2

**Review:**

Summary:

This paper addresses the computational aspects of Viterbi-based encoding for neural networks.

In usual Viterbi codes, input messages are encoded via a convolution with a codeword, and then decoded using a trellis. Now consider a codebook with n convolutional codes, of rate 1/k. Then a vector of length n is represented by inputing a message of length k and receiving n encoded bits. Then the memory footprint (in terms of messages) is reduced by rate k/n. This is the format that will be used to encode the row indices in a matrix, with n columns.  (The value of each nonzero is stored separately.)  However, it is clear that not all messages are possible, only those in the "range space" of my codes. (This part is previous work Lee 2018.)

The "Double Viterbi" (new contribution) refers to the storage of the nonzero values themselves. A weakness of CSR and CSC (carried over to the previous work) is that since each row may have a different number of nonzeros, then finding the value of any particular nonzero requires going through the list to find the right corresponding nonzero, a sequential task. Instead, m new Viterbi decompressers are included, where each row becomes (s_1*codeword_1 + s_2*codeword2 + ...) cdot mask, and the new scalar are the results of the linear combinations of the codewords.

Pros:
 - I think the work addressed here is important, and though the details are hard to parse and the new contributions seemingly small, it is important enough for practical performance.
 - The idea is theoretically sound and interesting.

Cons:
 - My biggest issue is that there is no clear evaluation of the runtime benefit of the second Viterbi decompressor. Compressability is evaluated, but that was already present in the previous work. Therefore the novel contribution of this paper over Lee 2018 is not clearly outlined.
 - It is extremely hard to follow what exactly is going on; I believe a few illustrative examples would help make the paper much clearer; in fact the idea is not that abstract.
 - Minor grammatical mistakes (missing "a" or "the" in front of some terms, suggest proofread.)

---

> ### Author Response · Authors · 2018-11-26
> **Response to AnonReviewer3**
>
> Thank you very much for the positive comments. We added the more experimental data of runtime analysis to address the Reviewer's main concern.
>
> Q1. My biggest issue is that there is no clear evaluation of the runtime benefit of the second Viterbi decompressor. Compressability is evaluated, but that was already present in the previous work. Therefore the novel contribution of this paper over [1] is not clearly outlined.
>
> We conducted additional simulations to evaluate the runtime benefit of the proposed method compared to that of the method in [1]. We generated random 512-by-512 matrices with pruning rate ranging from 70 % to 95 % and simulated the number of parameters fed to PEs in 10000 cycles. The assumptions used for the simulation and analysis data have been updated in Figure 6c of the revised manuscript. We could observe that proposed parallel weight decoding based on the second Viterbi decompressor allowed 10 % to 40 % more parameters to be fed to PEs than the previous design [1]. The proposed method outperformed both baseline method and [1] in all simulation results. Please note that the data described in Figure 6c has been updated from Figure 7, and our method shows better performance in new data compared to the data shown in the original manuscript. While preparing for the rebuttal, we realized that our simulation model did not fully exploit the parallelized weight and index decoding process of the proposed method. After further optimization, we could observe that the parameter feeding rate of the proposed method increased compared to the reported data in original manuscript. Therefore, we updated Figure 7 in original manuscript to Figure 6c in the updated manuscript according to the new data.
>
> Q2. It is extremely hard to follow what exactly is going on; I believe a few illustrative examples would help make the paper much clearer; in fact the idea is not that abstract.
>
> In the revision, we added the more precise mathematical description of the input and output of each block in Figure 1 and showed the change of the exact weight representation at each process. We first prune weights in a neural network with the Viterbi-based pruning scheme [1], then we quantize the pruned weights with the alternating quantization method [2]. Our main contribution is the third process, which includes encoding each weight with the Viterbi algorithm, and retraining for the recovery of accuracy. With our proposed method, the sparse and encoded weights are reconstructed to a dense matrix as described in Figure 2. Figure 2 illustrates the purpose of our proposed scheme, which is the parallelization of the whole sparse-to-dense conversion process with the VDs while maintaining the high compression rate.
>
> Q3. Minor grammatical mistakes (missing "a" or "the" in front of some terms, suggest proofread.)
>
> Thanks very much for the suggestions. We tried to fix grammatical mistakes as much as possible in the revision.
>
> Reference
> [1] Dongsoo Lee, Daehyun Ahn, Taesu Kim, Pierce I. Chuang, and Jae-Joon Kim. Viterbi-based pruning for sparse matrix with fixed and high index compression ratio. International Conference on Learning Representations (ICLR), 2018.
> [2] Chen Xu, Jianqiang Yao, Zouchen Lin, Wenwu Qu, Yuanbin Cao, Zhirong Wang, and Hongbin Zha. Alternating multi-bit quantization for recurrent neural networks. International Conference on Learning Representations (ICLR), 2018.

---

> > ### Comment · AnonReviewer3 · 2018-11-26
> > **Response to revision**
> >
> > Thanks for the revisions. Just to clarify, the plot (fig 6) is a rate, right? # param / xxx cycles? If not, it is a bit confusing; if so, that clarification would help, since rate is a more intuitive performance metric. Figure 1 is also great for added clarification.
> >
> > My only remaining suggestion is, if available, to have some runtime comparisons as well, accompanying fig 6, just as a more visible improvement metric. Otherwise, I think just clarifying what exactly Fig 6 is plotting already strengthens the paper significantly.

---

> > > ### Author Response · Authors · 2018-11-27
> > > **Follow-up response**
> > >
> > > Thank you very much for your quick and positive response. Considering your suggestion, we updated our manuscript as follows:
> > >
> > > To clarify what Fig. 6c is representing, we changed the label of Y axis of Fig. 6c to "Parameter feeding rate". Our data is describing the number of parameters fed to PEs in certain clocks - it is feeding rate, as you suggested.

---

### Official Review · AnonReviewer2 · 2018-11-01
**Can be an accept if they do more analyses**

**Rating:** 6
**Confidence:** 3

**Review:**

This paper presents a new way to represent a dense matrix in a compact format. First, the method prunes a dense matrix based on the Viterbi-based pruning. Then, the pruned matrix is quantized with alternating multi-bit quantization. Finally, the binary vectors produced by the quantization algorithm are further compressed with the Viterbi-based algorithm. It spots the problem of each existing approach and solve the problems by combining each method. The combination is new and the result is encouraging.

I find this paper is interesting and I like the strong results. It is an interesting combination of methods. However, the experiments are not enough to show that the proposed method is really needed to achieve the results. If these are answered well, I'd be happy to change my evaluation.

1. The method should be compared with other combinations of components. At least, it should be compared with "Multi-bit quantization only (Xu et al., 2018)" and "Multi-bit-quantization + Viterbi-based binary code encoding".

2. The experiments with "Don't Care" should go to the experiment section, and the end-to-end results should be present but not the ratio of incorrect bits.

3. Similarly, the paper will become stronger if it has some experimental results that compare quantization methods. In Section 3.3., it mentions that the conventional k-bit quantization was tried and significant accuracy drops were observed. I feel that this is a kind of things which support the proposed method if it is properly assessed.

4. When you say "slow" form something and propose a method to address it, I'd like to see some benchmark numbers. There is an experiment with simulation, but that does not seem to simulate the slow "sequential sparse matrix decoding process".

Minor comments:

* It was a bit hard to understand how a matrix is processed through the flowchart in Fig. 1 at first glance. It would help readers to understand it better if it has a corresponding figure which shows how a matrix is processed through the flowchart.

---

> ### Author Response · Authors · 2018-11-26
> **Response to AnonReviewer2**
>
> Thank you very much for the constructive comments. We tried to strengthen our claims by adding more experimental data which the Reviewer requested.
>
> 1. The proposed "Multi-bit-quantization + Viterbi-based binary code encoding" requires slightly larger memory footprint than "Multi-bit quantization only ([4])" because some of the Viterbi encoded bits have different indices from their corresponding quantization bits. Hence, the "Multi-bit quantization only" requires 10 % to 20 % smaller memory footprint than "Multi-bit-quantization + Viterbi-based binary code encoding" case. However, the main reason why we apply the Viterbi weight encoding is that parallel sparse-to-dense matrix conversion can be done by applying same Viterbi encoding process to the non-zero values and indices of the non-zero values in parallel. This parallel sparse-to-dense conversion makes the speed of feeding parameters to PEs 10 % to 40 % faster compared to [1] (Figure 6c).
>
> 2. Per Reviewer’s suggestion, the experimental results for the effectiveness of "Don’t Care" term have been moved to Section 4.1.
>
> 3. Per Reviewer's suggestion, we measured accuracy differences before and after Viterbi encoding for several quantization methods such as linear quantization ([2]), logarithmic quantization ([3]), and alternating quantization ([4]) methods with the same quantization bits (3-bit). The result shows that combination with alternating quantization and Viterbi weight encoding had only 2 % validation accuracy degradation after the Viterbi encoding was applied first right after the quantization and the accuracy was easily recovered with retraining. On the other hand, the combination with the other quantization methods and Viterbi weight encoding showed accuracy degradation as much as 71 %, which was too large to recover the accuracy with retraining. The accuracy difference mainly results from the uneven weight distribution. Because weights of neural networks usually are normally distributed, the composition ratio of '0' and '1' is not equal when the linear or logarithmic quantization is applied to the weights of neural networks. As we stated in the manuscript, Viterbi encoder tends to produce similar number of '0' and '1'. Therefore, we can conclude that under the same bit condition, alternating quantization method shows best accuracy and compatibility with our bit-by-bit Viterbi encoding scheme regardless of the type of neural networks.
>
> 4. We conducted additional simulations to compare sparse matrix reconstruction speed of [1] and the proposed method. We used a random 512-by-512 size matrix with various pruning rate ranging from 75 % to 95 %. We conducted the simulations under the assumptions described in Figure 6c. The simulation results are shown in Figure 6c in updated manuscript. We could observe that the proposed method could feed 10 % to 40 % more nonzero weights and input activations to PEs in same 10000 cycles compared to [1]. Proposed method could also feed parameters to PEs 20 % to 106 % faster compared to baseline method, which reads dense weight and activation matrices directly from DRAM. The improvement in the proposed scheme mainly comes from the parallelized process of assigning non-zero values to their corresponding indices in the weight matrix. While preparing addition data for the rebuttal, we realized that our simulation model did not fully exploit the parallelized weight and index decoding process of the proposed method. After further optimization, we could observe that the parameter feeding rate of the proposed method increased compared to the reported data in original manuscript. Therefore, we updated Figure 7 in original manuscript to Figure 6c in updated manuscript according to the new data.
>
> 5. We added the change of the exact weight representation at each process in Figure 1 to clarify the flowchart.
>
> Reference
> [1] Dongsoo Lee, Daehyun Ahn, Taesu Kim, Pierce I. Chuang, and Jae-Joon Kim. Viterbi-based pruning for sparse matrix with fixed and high index compression ratio. International Conference on Learning Representations (ICLR), 2018.
> [2] Darryl D. Lin, Sachin S. Talathi, and V. Sreekanth Annapureddy. Fixed point quantization of deep convolutional networks.  In Proceedings of the 33rd International Conference on International Conference on Machine Learning - Volume 48, ICML’16, pp. 2849–2858. 2016.
> [3] Daisuke Miyashita, Edward H. Lee, and Boris Murmann. Convolutional Neural Networks using Logarithmic Data Representation. CoRR, abs/1603.01025, 2016. URL https://arxiv.org/abs/1603.01025.
> [4] Chen Xu, Jianqiang Yao, Zouchen Lin, Wenwu Qu, Yuanbin Cao, Zhirong Wang, and Hongbin Zha. Alternating multi-bit quantization for recurrent neural networks. International Conference on Learning Representations (ICLR), 2018.

---

> > ### Comment · AnonReviewer2 · 2018-11-28
> > **Response to revision**
> >
> > Thank you for the revision. It looks better now.
> >
> > 1. I'd suggest to put what you wrote in the manuscript. You could just have additional rows in the tables and your consideration. You need to say that the additional cost is small enough for the benefit with some numbers.
> >
> > 2. The text says "To verify the effectiveness of using the "Don’t Care" elements, we apply our proposed method on the original network and pruned one. " Why don't you apply the proposed method on the pruned one with and without the "Don't Care" elements and compare the results?
> >
> > 3. I'd recommend to put the numbers in the manuscript. It could be added as a footnote.
> >
> > 4. Thank you for doing this.
> >
> > 5. Thanks.

---

> > > ### Author Response · Authors · 2018-11-30
> > > **Follow-up response**
> > >
> > > Thank you for your kind response. We tried to address your requests as below:
> > >
> > > 1. As suggested, we will add the information about the comparison between “Multi-bit quantization only” case and “Multi-bit-quantization + Viterbi-based binary code encoding" case in the manuscript when we are allowed to update the manuscript.
> > >
> > > 2. Thanks for the valuable comments. We actually initially considered applying the proposed method without “Don’t Care” elements on pruned networks. However, such a method has several issues and we decided to use the “Don’t Care” elements and the index matrix mask generated by the Viterbi-Pruning (1st step of the training process).
> > >
> > > First, applying proposed method on pruned networks without “Don’t Care” elements requires more bits per weight because at least two bits are required to represent each weight bit to express pruned bit as well as +1 and -1. Eg: +1,0(pruned bit),-1.
> > > Second, if the pruned bits have errors in the initial try, they need to be retrained as well as the non-zero bits during the retraining. As a result, another pruning needs to be done in each retraining step to maintain the pruning rate. Repeating the pruning in every retraining step in the loop leads to slower compression process and worse accuracy. Note that pruning is not repeated in our method (Fig. 1)
> > > Third, using “Don’t Care” elements helps to find good Viterbi encoded weight because Viterbi Decompressor produces “0” and “1” with 50% probability each. If “Don’t Care” is not used, the number of “1” and “0” in the weight matrix can be significantly different depending on the sparsity of the pruned weight matrix so it is hard to find the “good” weight matrix from the Viterbi Decompressor outputs.
> > >
> > > To verify, we conducted additional experiments on using the LSTM model on the PTB corpus applying the proposed method with and without “Don’t Care” elements. The test PPW of “without Don’t Care” case was 92.0, while the test PPW of “with Don’t Care” case was 84.6. We could observe that “Don’t Care” elements play significant role in improving the performance of the network compressed with proposed method. We will add this result to the manuscript to clarify that “Don’t Care” elements are required in the proposed method.
> > >
> > > 3. We will update the manuscript as suggested. Thank you very much.

---

### Official Review · AnonReviewer1 · 2018-11-01
**poorly written and fundamentally flawed**

**Rating:** 6
**Confidence:** 4

**Review:**

The paper proposes two additional steps to improve the compression of weights in deep neural networks. The first is to quantize the weights after pruning, and the second is to further encode the quantized weights.

There are several weaknesses in this paper. The first one is clarity. The paper is not very self-contained, and I have to constantly go back to Lee et al. and Xu et al. in order to read through the paper.

The paper can be made more mathematically precise. The input and output types of each block in Figure 1. should be clearly stated. For example, in Section 3.2, it can be made clear that the domain of the quantization function is the real and the codomain is a sequence k bits. Since the paper relies so heavily on Lee et al., the authors should make an effort to summarize the approach in a mathematically precise way.

The figures are almost useless, because the captions contain very little information. For example, the authors should at least say that the "D" in Figure 2. stands for delay, and the underline in Figure 4. indicates the bits that are not pruned. Many more can be said in all the figures.

The second weakness is experimental design. There are two conflicting qualities that need to be optimized--performance and compression rate. When optimizing the compression rate, it is important not to look at the test set error. If the compression rate is optimized on the test set, then the compressed model is nothing but a model overfit to the test set. The test set is typically small compared to the training set, so it is no surprise that the compression rate can be as high as 90%.

Optimizing compression rates should be done on the training set with a separate development set. The test set should not used before the best compression scheme is selected. Both the results on the development set and on the test set should be reported for the validity of the experiments. I do not see these experimental settings mentioned anywhere in the paper, and this is very concerning. Lee et al. seem to make similar mistakes, and it is likely that their experimental design is also flawed.

---

> ### Author Response · Authors · 2018-11-26
> **Response to AnonReviewer1**
>
> Thank you very much for the comments. We believe that this response can help the Reviewer to be more convinced about the validness of our experiments; in particular, the validness of our retraining methodology.
>
> Q1. The paper is not very self-contained, and I have to constantly go back to [1] and [2] in order to read through the paper.
>
> In the original manuscript, we had to limit the detailed information of the previous work due to the page limit. Based on the Reviewer’s comments, we added more description about the schemes we adopted from [1] and [2] in Appendix A.1 and A.2 of the revised manuscript.
>
> Q2. The input and output types of each block in Figure 1. should be clearly stated, and the figures are almost useless because the captions contain very little information.
>
> We tried to add more information to the figures in the revision. First, in Figure 1, we added the more mathematically precise description of input and output of each block to show how the exact weight representation is changed at each process. We also added additional explanation for 'D' of Figure 2 in its caption. For the Figure 4, we added the description of the underlined numbers.
>
> Q3. Optimizing compression rates should be done on the training set with a separate development set. The test set should not used before the best compression scheme is selected. Both the results on the development set and on the test set should be reported for the validity of the experiments.
>
> Thanks for pointing this out. We believe that this is the Reviewer 1's core question so would like to justify our results more in detail in this response and try to convince the Reviewer. We agree that optimizing compression rates should not use the test set before the best compression scheme is selected. In fact, in case of PTB and Wikitext-2 corpus, we already used the provided validation set and measured the test PPW only once after training (Table 2) in the original manuscript. From the Table 2, we can see that our proposed scheme maintains the accuracy of the uncompressed baseline network. On the other hand, the CIFAR-10 dataset does not include a separate validation set, so we had to use the test set in the retraining process. To avoid using the test set in the retraining process as the Reviewer pointed out, we randomly selected 5K validation images among the original 50K training images in CIFAR-10 dataset, and applied our scheme. Then, we observed the training and validation accuracy at each training epoch, and measured the test accuracy once after training. The accuracy results are as shown in the following table. Note the compression rates are the same as the data in Table 3 in the original manuscript.
>
> ----------------------------------------------------------------------------------
> Compression scheme   Validation Error (%)    Test Error (%)
> ------------------------------  ---------------------------   ---------------------
>           Baseline                             11.5                         12.2
>          Pruning [1]                         11.4                         12.2
>         VWM (Ours)                        11.4                         12.4
> ----------------------------------------------------------------------------------
>
> The test accuracy in the above table is about 1 % less than the accuracy which we reported in the originally submitted manuscript because the number of training data was decreased as part of the data set is used as a validation set.  However, the results show that our proposed method does not make the network be overfitted to test data as the Reviewer doubted because the difference between the accuracy for validation set and test set are consistent with the values from the previous works. Note that even the uncompressed baseline network exhibits similar accuracy difference between the validation error and the test error compared with the compressed networks. Therefore, we believe that our proposed compression method does not suffer from the concerned overfitting problem regardless of the types of neural networks or dataset.
>
> Reference
> [1] Dongsoo Lee, Daehyun Ahn, Taesu Kim, Pierce I. Chuang, and Jae-Joon Kim. Viterbi-based pruning for sparse matrix with fixed and high index compression ratio. International Conference on Learning Representations (ICLR), 2018.
> [2] Chen Xu, Jianqiang Yao, Zouchen Lin, Wenwu Qu, Yuanbin Cao, Zhirong Wang, and Hongbin Zha. Alternating multi-bit quantization for recurrent neural networks. International Conference on Learning Representations (ICLR), 2018.

---

> > ### Comment · AnonReviewer1 · 2018-11-26
> > **great update**
> >
> > Thanks for the update. The paper now reads better. Since the Viterbi pruning is heavily used in the paper, I still think it deserves more text in section 3.1. I also think the captions can include more information, such as Figure 3, 4, and 5.
> >
> > The numbers provided above are good and should be included in the paper. The authors should also explain clearly how the experiments are carried out. It would be great to provide the unpruned results in all tables.
> >
> > The experimental methodology should be strictly followed for all experiments, i.e., using a data set from the training set (subsampled or not) to prune the network, tuning the hyperparameters on the validation set, and testing the models on the test set. It is especially important since the pipeline involves a re-training step. We do not tolerate any hyperparameter tuning on the test set, and any paper that does this should be rejected.
> >
> > The numbers haven't been changed. I will raise my score once the experiments are done properly.

---

> > > ### Author Response · Authors · 2018-11-27
> > > **Follow-up response**
> > >
> > > Thank you very much for your positive and rapid response. We appreciate it. As suggested, we made the following updates in the manuscript.
> > >
> > > 1.   We included the new numbers which we showed in the previous response in the manuscript. The unpruned results have been also added in all the tables. As a result, Table 1, Table 2, Table 3 have been updated. In particular, table 3 now shows the accuracy data for validation set and the test set separately.
> > > 2.   At the bottom of Section 3.4, we explicitly stated that we did not use the test set for training.
> > > 3.   In section 4.2, we explained that we randomly selected 5K validation set to avoid using the test set for training.
> > > 4.   In section 3.1, we added a brief sentence for describing Viterbi-pruning more. We agree that Viterbi-pruning deserves more description in the manuscript but please understand that the page limit still makes us to defer the most of detail to the appendix A.1.
> > > 5.   We added more information in the captions for Fig. 3, 4. 5.

---

> > > > ### Comment · AnonReviewer1 · 2018-11-27
> > > > **LM and MT experiments?**
> > > >
> > > > What about the LM and MT experiments? Is the pruning also done on a subset of the training set and tuned on the validation set? If so, please state them clearly in the paper. If not, the authors should redo those experiments as well.
> > > >
> > > > When revising the captions, please add useful descriptions not just more words. For example, please describe what the nodes, arrows, and numbers mean. Please don't just revise the captions in figure 5. Apply this to all captions.

---

> > > > > ### Author Response · Authors · 2018-11-28
> > > > > **Clarification on LM and MT cases**
> > > > >
> > > > > The language modeling and machine translation experiments also did not use test set in the training. The pruning was done on the training set and tuned on validation set. The Table A1,A2,A3,A4 already have separate accuracy data for validation error and test error.
> > > > > Unfortunately, we cannot upload the revised manuscript any more. We will update the manuscript as follows when we are allowed to revise it again.
> > > > > At the bottom of Section 3.4 of the current manuscript, there is a sentence which we intended to state that we did not use the test set for hyperparameter tuning throughout the paper.  The sentence is that “Note that entire training process used the training dataset and the validation dataset only to decide the best compressed weight data. The accuracy measurement for the test dataset was done only after training is finished so that any hyperparameter was not tuned on the test dataset.”.
> > > > > For further clarification, we will add the following sentence right after it.
> > > > > “All the experiments in this paper followed the above training principle”.
> > > > > We will also add the following sentences in appendix A.3 and A.4 to make it clear.
> > > > > “As described in Section 3.4, the compression is done on the training set and tuned on validation set so that any hyperparameter was not tuned on the test dataset during the process.”
> > > > >
> > > > > In fact, the information you requested for Fig. 5 was already included in the figure itself. In the next revision, we will elaborate the information in the caption for better readability as follows.
> > > > > “In the figure, each circle indicates a state. A circle which is the source point of arrows indicates a current state and a circuit which is the sink point of arrows indicates a next state. The arrow indicates a transition from the current state to the next state. Depending on the input to VD, each current state can be switched to one of the two potential next states in the next clock. The number in a circle indicates the index for the state.”
> > > > > For Fig. 2, we will add the following sentence in addition to the existing description in the caption.
> > > > > “the + symbol indicates an adder.”
> > > > > Captions for Figs. 1, 3, 4, 6 have been already updated with additional information. We will be happy to elaborate further if the Reviewer has additional suggestions.

---

> > > > > > ### Comment · AnonReviewer1 · 2018-11-28
> > > > > > **score revision**
> > > > > >
> > > > > > Thanks. I missed that sentence in section 3.4. I have revised the score. The runtime issue that other reviewers had is a tough one. I will let other reviewers lead the discussion. Thanks for the great work.

---

> > > > > > > ### Author Response · Authors · 2018-11-28
> > > > > > > **Thank you very much!**
> > > > > > >
> > > > > > > Thank you very much for your constructive feedback and score revision. We appreciate it!

---

### Meta-Review · Area_Chair1 · 2018-12-14
**Efficient weight encoding which is important for a practical standpoint**

**Confidence:** 4
**Recommendation:** Accept (Poster)

**Metareview:**

The authors propose an efficient scheme for encoding sparse matrices which allow weights to be compressed efficiently. At the same time, the proposed scheme allows for fast parallelizable decompression into a dense matrix using Viterbi-based pruning.
The reviewers noted that the techniques address an important problem relevant to deploying neural networks on resource-constrained platforms, and although the work builds on previous work, it is important from a practical standpoint.
The reviewers noted a number of concerns on the initial draft of this work related to the experimental methodology and the absence of runtime comparison against the baseline, which the reviewers have since fixed in the revised draft. The reviewers were unanimous in recommending that the revision be accepted, and the authors are requested to incorporate the final changes that they said they would make in the camera-ready version.